# Comparative Multicentric Evaluation of Inter-Observer Variability in Manual and Automatic Segmentation of Neuroblastic Tumors in Magnetic Resonance Images

**DOI:** 10.3390/cancers14153648

**Published:** 2022-07-27

**Authors:** Diana Veiga-Canuto, Leonor Cerdà-Alberich, Cinta Sangüesa Nebot, Blanca Martínez de las Heras, Ulrike Pötschger, Michela Gabelloni, José Miguel Carot Sierra, Sabine Taschner-Mandl, Vanessa Düster, Adela Cañete, Ruth Ladenstein, Emanuele Neri, Luis Martí-Bonmatí

**Affiliations:** 1Grupo de Investigación Biomédica en Imagen, Instituto de Investigación Sanitaria La Fe, Avenida Fernando Abril Martorell, 106 Torre A 7planta, 46026 Valencia, Spain; leonor_cerda@iislafe.es (L.C.-A.); marti_lui@gva.es (L.M.-B.); 2Área Clínica de Imagen Médica, Hospital Universitario y Politécnico La Fe, Avenida Fernando Abril Martorell, 106 Torre A 7planta, 46026 Valencia, Spain; sanguesa_cin@gva.es; 3Unidad de Oncohematología Pediátrica, Hospital Universitario y Politécnico La Fe, Avenida Fernando Abril Martorell, 106 Torre A 7planta, 46026 Valencia, Spain; blanca_martinez@iislafe.es (B.M.d.l.H.); canyete_ade@gva.es (A.C.); 4St. Anna Children’s Cancer Research Institute, Zimmermannplatz 10, 1090 Vienna, Austria; ulrike.poetschger@ccri.at (U.P.); sabine.taschner@ccri.at (S.T.-M.); vanessa.duester@ccri.at (V.D.); ruth.ladenstein@ccri.at (R.L.); 5Academic Radiology, Department of Translational Research, University of Pisa, Via Roma, 67, 56126 Pisa, Italy; michela.gabelloni@med.unipi.it (M.G.); emanueleneri1@gmail.com (E.N.); 6Departamento de Estadística e Investigación Operativa Aplicadas y Calidad, Universitat Politècnica de València, Camí de Vera s/n, 46022 Valencia, Spain; jcarot@eio.upv.es

**Keywords:** tumor segmentation, neuroblastic tumors, deep learning, manual segmentation, automatic segmentation, inter-observer variability

## Abstract

**Simple Summary:**

Tumor segmentation is a key step in oncologic imaging processing and is a time-consuming process usually performed manually by radiologists. To facilitate it, there is growing interest in applying deep-learning segmentation algorithms. Thus, we explore the variability between two observers performing manual segmentation and use the state-of-the-art deep learning architecture nnU-Net to develop a model to detect and segment neuroblastic tumors on MR images. We were able to show that the variability between nnU-Net and manual segmentation is similar to the inter-observer variability in manual segmentation. Furthermore, we compared the time needed to manually segment the tumors from scratch with the time required for the automatic model to segment the same cases, with posterior human validation with manual adjustment when needed.

**Abstract:**

Tumor segmentation is one of the key steps in imaging processing. The goals of this study were to assess the inter-observer variability in manual segmentation of neuroblastic tumors and to analyze whether the state-of-the-art deep learning architecture nnU-Net can provide a robust solution to detect and segment tumors on MR images. A retrospective multicenter study of 132 patients with neuroblastic tumors was performed. Dice Similarity Coefficient (DSC) and Area Under the Receiver Operating Characteristic Curve (AUC ROC) were used to compare segmentation sets. Two more metrics were elaborated to understand the direction of the errors: the modified version of False Positive (FPRm) and False Negative (FNR) rates. Two radiologists manually segmented 46 tumors and a comparative study was performed. nnU-Net was trained-tuned with 106 cases divided into five balanced folds to perform cross-validation. The five resulting models were used as an ensemble solution to measure training (n = 106) and validation (n = 26) performance, independently. The time needed by the model to automatically segment 20 cases was compared to the time required for manual segmentation. The median DSC for manual segmentation sets was 0.969 (±0.032 IQR). The median DSC for the automatic tool was 0.965 (±0.018 IQR). The automatic segmentation model achieved a better performance regarding the FPRm. MR images segmentation variability is similar between radiologists and nnU-Net. Time leverage when using the automatic model with posterior visual validation and manual adjustment corresponds to 92.8%.

## 1. Introduction

Neuroblastic tumors are the most frequent extracranial solid cancers in children. They comprise ganglioneuroma, ganglioneuroblastoma and neuroblastoma. Ganglioneuroma is composed of gangliocytes and mature stroma and is the most benign. Ganglioneuroblastoma is formed by mature gangliocytes and immature neuroblasts and has an intermediate malignant potential [1]. The most frequent type is neuroblastoma, which is more immature and undifferentiated. It is a heterogeneous neoplasm that shows different behavior based on biological, clinical and prognostic features, some tumors undergo spontaneous regression, while others progress with fatal outcomes despite therapy [2]. Neuroblastic tumors show a wide range of variability in their position. The most common sites of origin of neuroblastic tumors are the adrenal region (48%), extra-adrenal retroperitoneum (25%) and the chest (16%), followed by the neck (3%) and the pelvis (3%) [3]. Furthermore, they show high variability in their size, shape and boundaries, resulting in a common challenging task to differentiate them from the neighboring structures.

Tumor diagnosis, prognosis and the decision on respective treatment/disease management are mainly based on information obtained from imaging, including magnetic resonance (MR) [4]. Additionally, multiparametric data, radiomic features and imaging biomarkers, can provide the clinician with relevant information for disease diagnosis, characterization, and evaluation of aggressiveness and treatment response [5].

In order to ensure the best usability of imaging, it is essential to develop a robust and reproducible imaging processing pipeline. One of the most relevant steps involves segmentation, which consists of placing a Region of Interest (ROI) on a specific area (e.g., a tumor), with the assignment and labeling of voxels in the image that correspond to the ROI. Tumor segmentation can be performed in three different ways: manual, semiautomatic and automatic. Manual segmentation is usually performed by an experienced radiologist. This is usually done slice-by-slice, but is also possible in 3D, with the expert either encircling the tumor or annotating the voxels of interest. This is a reliable but time-consuming method that hinders the radiologists’ workflow, especially in cases of mass data processing. However, manual segmentation is observer-dependent and may show wide inter and intra-observer variability [6,7]. This variability is influenced by some objective factors, such as organ/tumor characteristics or contour, and by some subjective factors related to the observer, such as their expertise or coordination skills [6].

Semiautomatic segmentation tries to solve some of the problems related to manual segmentation [8]. By assisting the segmentation with algorithms, for example, by growing the segmentation over a region or expanding the segmentation to other slices to eliminate the need for a slice-by-slice segmentation, the effort and time required from the user can be reduced. However, inter-observer variability is still present, as the manual part of the segmentation and the settings of the algorithm influence the result.

In the case of neuroblastic tumors, several studies have explored the development of semiautomatic segmentation algorithms. They have been performed on Computed Tomography (CT) or MR images, making use of mathematical morphology, fuzzy connectivity and other imaging processing tools [9,10,11]. However, they have included a very low number of cases and the findings show little improvement with respect to manual approaches. To the best of our knowledge, a robust and generalizable solution for neuroblastoma segmentation has not been yet devised.

Nowadays, most advanced tools are built to be used as automatic segmentation methods, which, by definition, do not rely on user interaction. These solutions are built with deep-learning segmentation algorithms [12], usually based on convolutional neural networks (CNNs). CNNs use several sequential convolution and pooling operations in which images are processed to extract features and recognize patterns using the image itself to be trained during the learning process [13]. One of the most commonly used is the U-Net architecture, consisting of a contracting path to capture context and a symmetric expanding path that enables precise localization, which achieves very good performance in segmentation of different types of cancer [14,15]. Nevertheless, its applicability to specific image analysis and its reproducibility in different structures or lesions has been observed to be limited [16].

Recently, a new solution based on CNNs algorithms called nnU-Net has been proposed. It consists of an automatic deep learning-based segmentation framework that automatically configures itself, including preprocessing, network architecture, training and post-processing, and adapts to any new dataset, surpassing most existing approaches [16,17].

The aim of this study was to assess the inter-observer variability in manual and automatic segmentation of neuroblastic tumors. We hypothesize that the state-of-the-art deep learning framework nnU-Net can be used to automatically detect and segment neuroblastic tumors on MR images, providing a more robust, universal and error-free solution than that obtained by the manual segmentation process. This comparison is performed by evaluating the inter-observer variability between two radiologists. The automatic segmentation model is trained, fine-tuned and validated with cases from different European institutions and then compared to manual segmentation. Previous expert tumor delineation is performed as there does not exist an open-access annotated data set dedicated to this specific tumor.

The automatic segmentation model is then applied to a group of patients from the training set. The time needed for the automatic segmentation (with manual adjustment when necessary) is compared to the time required to manually segment the same cases from scratch.

## 2. Materials and Methods

### 2.1. Participants

A retrospective multicenter and international collection of 132 pediatric patients with neuroblastic tumors who had undergone a diagnostic MR examination was conducted.

All patients had received a diagnosis of neuroblastic tumor with pathological confirmation between 2002 and 2021. Patients and MR data were retrospectively obtained from 3 centers in Spain (n = 73, La Fe University and Polytechnic Hospital, including 21 cases from European Low and Intermediate Risk Neuroblastoma Protocol clinical trial (LINES)), Austria (n = 57, Children’s Cancer Research Institute from SIOPEN High Risk Neuroblastoma Study (HR-NBL1/SIOPEN) current accrual over 3000 patients from 12 countries), and Italy (n = 4, Pisa University Hospital). The study had the corresponding institutional Ethics Committee approvals from all involved institutions. This data set was collected within the scope of PRIMAGE (PRedictive In-silico Multiscale Analytics to support cancer personalized diaGnosis and prognosis, empowered by imaging biomarkers) project [5]. Age at first diagnosis was 37.6 ± 39.3 months (mean ± standard deviation, range 0 to 252 months, median of 24.5 months ± 54 interquartile range (IQR)), with a slight female predominance (70 females, 62 males).

Histology of the tumor was neuroblastoma (104 cases), ganglioneuroblastoma (18 cases) and ganglioneuroma (10 cases). Tumor location was classified as abdominopelvic (105 cases, 59 of them from the adrenal gland, 32 with abdominal non-adrenal location and 14 with a pelvic location) or cervicothoracic (27 cases, 18 of them thoracic, 2 with an exclusive cervical location and 7 affecting both thoracic and cervical regions).

Imaging data from SIOPEN clinical trials (HR-NBL1 and LINES) were collected and centrally stored on an Image Management Server maintained by the Austrian Institute of Technology (AIT) in order to be properly pseudonymized with the European Unified Patient Identity Management (EUPID) [18] system enabling a privacy-preserving record linkage and a secure data transition to the PRIMAGE context. Other imaging data not coming from a SIOPEN trial received a EUPID pseudonym through the direct upload to the PRIMAGE platform. All collected images have been stored in the PRIMAGE platform to be used for further investigation.

The MR images accounted for a high data acquisition variability, including different scanners, vendors and protocols, from the different institutions. MR images were acquired with either a 1.5 T (n = 116) or 3 T (n = 16) scanner, manufactured by either General Electric Healthcare (Signa Excite HDxt, Signa Explorer) (n = 51), Siemens Medical (Aera, Skyra, Symphony, Avanto) (n = 54) or Philips Healthcare (Intera, Achieva, Ingenia) (n = 27). The MR protocol varied among the institutions. Essentially, MR studies consisted of T1-weighted (T1W), T2- weighted (T2w) and/or T2w with fat suppression (T2w fat-sat), Diffusion-weighted (DW) and Dynamic Contrast-enhanced (CET). Chest images were acquired with respiratory synchronization. Mean FOV size was 410 mm, and median FOV was 440 mm (range of 225 to 500 mm).

### 2.2. Manual Image Labeling

Tumor segmentation was performed on the transversal T2w fat-sat images as they yield the maximum contrast between the tumor and the surrounding organs (48 cases). T2w images were used when T2w fat-sat images were not available (84 cases). All images were obtained in DICOM format. The open source ITK-SNAP (version 3.8.0) (www.itksnap.org) tool [19] was used for the manual tumor segmentation by two radiologists (with 30 (Radiologist 1) and 5 (Radiologist 2) years of experience in pediatric radiology, respectively) with prior experience with manual segmentation tools. All the tumors (132 cases) were manually segmented by Radiologist 2. For the inter-observer variability study, 46 cases were independently segmented by both radiologists after agreement on the best tumor definition criteria. To increase reproducibility, a restrictive segmentation methodology was established, excluding doubtful peripheral areas. If the tumor contacted or encased a vessel, the vessel was excluded from the segmentation. When the tumor infiltrated neighboring organs with ill-defined margins, the DWI and CE images were reviewed to exclude non-tumoral areas. Lymph nodes separated from the tumor and metastases were also excluded. Each of the readers performed a blinded annotation of all the cases independently. Finally, the obtained segmentation masks were exported in NIfTI format (nii) and were considered the ground truth ROIs.

Tumor volume was obtained from the 132 masks performed by Radiologist 2. The median volume of all the masks was 116,518 mm^3^ (±219,084 IQR) and the mean volume was 193,634 mm^3^.

### 2.3. Study Design and Data Partitioning

Our study consisted of two parts (Figure 1). Firstly, the inter-observer variability in manual MR segmentation was analyzed by comparing the performance of two radiologists in 46 cases of neuroblastic tumor. Secondly, the training and validation of the automatic segmentation model based on nnU-Net architecture were performed, dividing the dataset into two cohorts: training-tuning and validation. A balanced and stratified split of the cases from both cohorts was implemented to eliminate sampling bias and to guarantee the heterogeneity of both datasets in order to construct a reproducible and universal model. Stratified sampling with the scikit-learn library [20] was used, considering four variables: manufacturer (Siemens/Philips/GE), magnetic field strength (1.5 T/3 T), tumor location (abdominopelvic/cervicothoracic) and segmented sequence (T2w/T2w fat-sat). (Table 1).

A first cohort (80% of cases, n = 106) was selected to train and fine-tune the model with a 5-fold cross-validation approach. A second cohort (20% of patients, n = 26) was used for validation.

### 2.4. Convolutional Neural Network Architecture

The automatic segmentation model was developed using the state-of-the-art, self-configuring framework for medical segmentation, *nnU-Net* [16]. All the images were resampled with a new voxel spacing: [z, x, y] = [8, 0.695, 0.695], corresponding to the average values within the training data set. The model training was performed along 1000 epochs with 250 iterations each and a batch size of 2. The loss function to optimize each iteration was based on the Dice Similarity Coefficient (DSC). A z-score normalization was applied to the images.

The model employed a 3D net and was trained with a cross-validation strategy, which is a statistical technique frequently used to estimate the skill of a machine learning model on unseen data [21]. The training-tuning dataset (n = 106) was partitioned into 5 subsets or folds of 21 or 22 non-overlapping cases each. Each of the 5 folds was given an opportunity to be used as a held-back test set, whilst all other folds collectively were used as a training dataset. A total of 5 models were fit and evaluated on the 5 hold-out test sets and performance metrics were reported (median and IQR were reported as the distribution of the results was not normal in all the cases. Confidence interval (CI) was also calculated).

Additionally, the 5 resulting segmentation models obtained using the cross-validation method were used as an ensemble solution to test all the cases of the training-tuning (n = 106) and the validation (n = 26) data sets in order to measure training and validation performance independently.

### 2.5. Analysis and Metrics

To compare segmentation results, different metrics have been described in the literature. The main metric used in this study for the evaluation of results was the DSC [22], a spatial overlap index and a reproducibility validation metric [23]. Its value can range from 0 (meaning no spatial overlap between two sets) to 1 (indicating complete overlap). DSC index has been widely used to calculate the overlap metric between the results of segmentation and ground truth, and is defined as [24,25]:$$\mathrm{DSC}=\frac{2\mathrm{TP}}{2\mathrm{TP}+\mathrm{FP}+\mathrm{FN}}$$

The ROC AUC metric was also calculated. The ROC curve, as a plot of sensitivity against 1-specificity, normally assumes more than one measurement. For the case where a test segmentation is compared to a ground truth segmentation, we consider a definition of the AUC as [26]:$$\mathrm{AUC}=1-\frac{1}{2}\left(\frac{\mathrm{FP}}{\mathrm{FP}+\mathrm{TN}}+\frac{\mathrm{FN}}{\mathrm{FN}+\mathrm{TP}}\right)$$

Since metrics have different properties, selecting a suitable one is not a trivial task, and therefore a wide range of metrics have been previously developed and implemented to approach 3D image segmentation [26]. For our study, two spatial overlap-based metrics were specifically designed to gain a deeper understanding of the direction of the errors encountered: the false positive (FP) and false negative (FN) rates, independently, with respect to the ground truth, which consisted of the manual segmentation performed by Radiologist 2 (Figure 2).

The rate of FP of the automatic segmentation to the ground truth (modified version of FPR) considered those voxels that were identified by the net as tumor but corresponded to other structures, divided by the voxels that actually corresponded to the ground truth mask (TP + FN voxels). This definition differs from the FPR used in standard statistical problems in the exclusion of the true negative (TN) term from the mathematical expression, as the TN voxels correspond to the image background in a segmentation task and not to the segmentation masks intended to be compared.
$$\mathrm{FPRm}=\frac{\mathrm{FP}}{\mathrm{TP}+\mathrm{FN}}$$

The rate of FN of the automatic segmentation to the ground truth (FNR) measured voxels belonging to the tumor that the net did not include as such, divided by the ground truth voxels.
$$\mathrm{FNR}=\frac{\mathrm{FN}}{\mathrm{TP}+\mathrm{FN}}=1-\;\mathrm{Sensitivity}\;\mathrm{or}\;\mathrm{Recall}$$

For consistency reasons and to facilitate the understanding of the results, the FPRm and FNR metrics are reported as 1-self, resulting in a maximum of 1 for a complete voxel-wise agreement and a minimum of 0 for a null similitude.

### 2.6. Time Sparing

For comparing the time leverage, the final automatic segmentation model was applied to 20 cases from the training set, corresponding to 4 cases per fold to account for the heterogeneity of the data set. Cases were independently segmented manually from scratch by Radiologist 2, and the mean time (in minutes) necessary to perform that task was compared to the mean time required to obtain the masks with the automatic model. As some variability may exist in the final automatic masks, a human-based validation by Radiologist 2 was performed, and the mean time required to visually validate and manually edit the resulting automatic masks (when needed) was compared to the time necessary to manually segment them from scratch. To remove a potential software-related bias, the open source ITK-SNAP tool [19] was used for both manual and automatic correction approaches.

## 3. Results

### 3.1. Inter-Observer Comparison for Manual Segmentation

The segmentation results obtained by Radiologist 1 were compared to those of Radiologist 2 to measure inter-observer variability (Table 2). The median DSC was found to be 0.969 (±0.032 IQR). The median FPRm was 0.939 (±0.063 IQR), resulting in a high concordance between both radiologists according to the non-tumor included voxels. The median FNR was 0.998 (±0.008 IQR), meaning that Radiologist 1 did not miss tumor during the segmentation. AUC ROC was 0.998 (Figure 3).

### 3.2. Comparison between Radiologist and nnU-Net

As the 106 cases of the training group were divided into five folds of 21 or 22 cases to perform cross-validation, each fold achieved different DSC results (Table 3) (Figure 4 and Figure 5).

Of the 106 cases, 27 had a DSC value <0.8: folds 0, 2 and 4 had 6 cases each; folds 1 and 3 had 5 cases each. The mean age for these cases was 32.7 ± 30.3 months and the median age was 19.8 months. They had a median volume of 75,733 mm^3^ (±42,882 IQR).

From these 27 cases, 8 had a DSC from 0 to 0.19, being in all the cases < 0.01; 3 cases had a DSC ≥ 0.2 to 0.39; 1 case had a DSC ≥ 0.4 to 0.59; and 11 cases had a DSC ≥ 0.6 to 0.8. Cases showing high variability (DSC < 0.8) after automatic segmentation were analyzed by Radiologist 2 to identify the reasons for the low level of agreement. Regarding the eight cases with DSC < 0.01, the net had segmented extensive lymph nodes instead of the primary tumor in three cases. In another two cases, the net segmented other structures instead of the tumor (gallbladder or left kidney). In another three cases, the net did not identify any structure from the original DICOM and thus did not perform any mask.

Of the remaining 19 cases with a DSC < 0.8, 18 cases showed differences as the net localized the tumor well but did not completely segment it or presented variability in the borders, especially in cases with surrounding lymph nodes. One case had bilateral tumors and the net only detected one of them.

Posteriorly, the five resulting segmentation models obtained using the cross-validation method were used as an ensemble solution to test all the cases of the training-tuning (n = 106). We obtained a median DSC of 0.965 (±0.018 IQR) and AUC ROC of 0.981. The FPRm for this ensemble solution was 0.968, and FNR was 0.963. For comparing means of DSC attending to the effects of location (abdominopelvic or cervicothoracic) and magnetic field strength (1.5 or 3 T) (Table 4), an ANOVA test was performed, showing that there were no differences in DSC mean values for the location and magnetic field factors, and the results repeated after considering atypical values and removing them.

When introducing age and volume as corrective factors in the evaluation of the effects of location and magnetic field in the DICE, no differences were observed in the results of the analyses. Age and volume have no significant effect and do not show any trend in the DICE (*p*-value = 0.052 for age and 0.169 for volume). Therefore, the effects of location and magnetic field, as well as their interaction, continue to be insignificant when the correction for age and volume is introduced.

Focusing on the direction of the errors between both sets (ground truth vs. automatic segmentation), the median FPRm is 0.968 (±0.015 IQR), meaning that the mask is including as tumor 3.2% of voxels that are not included in the ground truth. The median FNR is 0.963 (±0.021 IQR), so the automatic tool does not include 3.7% of the voxels included in the ground truth mask.

### 3.3. Validation

The validation was performed at the end of the model development to test for model overfitting which could result in an overestimation of the model performance. The median DSC result for validation was 0.918 (±0.067 IQR) and AUC ROC was 0.968 (Table 5) (Figure 5).

Of the 26 cases in the validation dataset, 4 had a DSC value < 0.8: 3 cases had a DSC ≥ 0.4 to 0.59; and 1 case had a DSC ≥ 0.6 to 0.8. These cases were analyzed by Radiologist 2 to identify the reasons for the low level of agreement. Regarding the three cases with DSC <0.6, the net had segmented extensive lymph nodes besides the primary tumor in two cases, and identified only a part of the tumor in one case. In the case with a DSC ≥ 0.6 to 0.8, the net segmented lymph nodes besides the primary tumor.

To compare the validation results to the inter-radiologist agreement, Radiologist 1 manually segmented the cases from the validation dataset. We compared the segmentation of Radiologist 1 to the segmentations of Radiologist 2 and the automatic model (Table 5).

For comparing the time leverage, we performed a comparison of the mean time needed to manually segment 20 cases (418 slices) from scratch with the mean time required by the automatic model to segment them. Cases segmented manually required a mean time of 56 min per case, while the mean time needed to obtain each mask with the automatic model was 10 s (0.167 min), resulting in a time reduction of 99.7%.

As some variability may exist in the final automatic masks, a human-based visual validation of the masks was performed. All the segmentations were visually validated, and manual editing and adjustment of the automatic masks were performed when needed (12 cases were edited, including 92 slices). The mean time to perform these processes was 4.08 min (±2.35 SD) and the median time was 4 min. This was compared to the time necessary to manually segment the masks from scratch, showing a time reduction of 92.8%.

## 4. Discussion

The inter-observer variability when performing manual segmentation of neuroblastic tumors in T2w MR images indicates that there is a high concordance between observers (median DSC overlap index of 0.969 (±0.032 IQR)). The discrepancies between observers may be due to the heterogeneous nature of the neuroblastic tumors and to the intrinsic variability of the manual segmentation related to individual skills and level of attention to detail. In our study, both radiologists are pediatric radiologists with previous experience in segmentation tasks. Radiologist 2 was considered the ground truth for practical reasons, as the segmentation of the whole dataset had been performed by this observer. Nevertheless, the manual ground truth mask may itself have some errors that are intrinsically associated with the human-based segmentation methodology. Joskowicz et al [6] investigated the variability in manual delineations on CT for liver tumors, lung tumors, kidney contours and brain hematomas between 11 observers, and concluded that inter-observer variability is large and even two or three observers may not be sufficient to establish the full range of inter-observer variability. Montagne et al [27] compared the inter-observer variability for prostate segmentation on MR performed by seven observers and concluded that variability is influenced by changes in prostate morphology. Therefore, delineation volume overlap variability for different structures and observers is large [28].

In our study, expert tumor delineation performed as the best (although not perfect) approach to truth. The evaluation of the voxel-wise similarity between the ground truth and the automatically segmented mask demonstrates that the state-of-the-art deep learning architecture nnU-Net can be used to detect and segment neuroblastic tumors on MR images, with a median DSC of 0.965 (±0.018 IQR), achieving a strong performance and surpassing the methods and results obtained in previous studies that approached the problem of neuroblastoma segmentation [9,10,11]. However, no previous literature has demonstrated the performance of a CNN-based solution in neuroblastic tumor. nnU-Net sets a new state-of-the-art in various semantic segmentation challenges and displays strong generalization characteristics for other structures [16,17]. Our results suggest that this automatic segmentation tool introduces a variability equivalent to that observed in the manual segmentation process in neuroblastic tumors. Previous studies related to breast tumors showed that segmentation algorithms may improve manual variability [29].

When analyzing the direction of the errors in a tumor segmentation problem, our recommendation is to give more relevance to the FPRm, aiming to minimize the included FP voxels with respect to the ground truth, as this metric represents those voxels that belong to adjacent organs or structures, which could introduce a strong bias in the extraction of quantitative imaging features for the development of radiomics models. The influence of FN in radiomics models seems less important, as it may not have a significant impact if some peripheral tumor voxels are missed. The effect of manual inter-observer segmentation variability on MR-based radiomics feature robustness has been described previously in other tumors such as breast cancer [30].

When assessing the FPRm and FNR between the manual segmentations performed by the two radiologists, the median FPRm is 0.939 (±0.063 IQR), indicating that 6.1% of the voxels were misclassified as tumor, while the median FNR is 0.998 (±0.008 IQR), therefore, the manual segmentation of Radiologist 1 did not include 0.2% of voxels included in the ground truth mask. Regarding manual ground truth vs. automatic segmentation, we observe that the median FPRm is 0.968 (±0.015 IQR). Therefore, the automatic segmentation tool generates masks with an average of 3.2% non-tumoral voxels. The median FNR corresponds to a value of 0.963 (±0.021 IQR), therefore, the automatic tool fails to include 3.7% of tumoral voxels. The results obtained demonstrate that the automatic segmentation model achieves a better performance regarding the FPRm, which is a great advantage in segmentation tasks for the posterior extraction of quantitative imaging features.

With regards to the time required for the segmentation process, an average time reduction of 99.7% was obtained when comparing the automatic model with the manual segmentation methodology. As some errors and variability may exist in the final automatic masks, this result is over-optimistic, as in practice the reader has to visually validate the quality of all the segmentations provided by the automatic tool before introducing some corrections, if needed. A human-based visual validation of the masks is recommended to edit and adjust the automatic masks. In our study, this process of validation and correction of the automatic masks that needed adjustment reduced the time required for segmentation from scratch by 92.8%. Correction time was influenced by the intrinsic difficulty of segmentation of each tumor, as there were tumors easier to segment (e.g., more homogeneous, with sharper margins, without lymph nodes) that did not need corrections or required slight adjustments, while there were more challenging tumors with contrast variations close to organ borders and of similar appearance to surrounding structures that required more time to be corrected. Overall, the application of the automatic model results in a great leverage of the time required to perform the segmentation process, facilitating the workflow for radiologists.

In addition, the human-based analysis of the masks performed by the net is useful to gain insights and correct for potential human mistakes and biases/outliers within the data set, which could be used to retrain the model, increasing the model’s overall accuracy and robustness.

There are some limitations to this study. Segmentations were performed only by two observers, so it may only represent a fraction of the full range of inter-observer variability and may not be enough to establish a reference standard. Furthermore, both were experienced radiologists, as previous medical knowledge and expertise are assumed to contour highly heterogeneous neuroblastic tumors. Therefore, manual segmentations performed by other users (less experienced radiologists, other clinical users, non-medical staff) are expected to encounter higher inter-observer variability. Another potential limitation is that tumors may associate extensive lymph nodes or can present contact with them, making their differentiation difficult in some cases, which, as we have proved, can lead to errors in the segmentation performed by the net. Finally, as has been pointed out, the mask that is considered to be the ground truth may itself have some errors that are associated intrinsically with the manual segmentation process, and the comparison of the nnU-Net results was performed with the segmentations done by a single radiologist.

## 5. Conclusions

MR image segmentation accuracy of neuroblastic tumors is observed to be comparable between radiologists and the state-of-the-art deep learning architecture nnU-Net. The automatic segmentation model achieves a better performance regarding the FPRm, which is a great advantage in segmentation tasks for the posterior extraction of quantitative imaging features. Moreover, the time leverage when using the automatic model corresponds to 99.7%. A human-based validation based on manual editing of the automatic masks is recommended and corresponds to a reduction of time of 92.8% compared to the fully manual approach, reducing the radiologist’s involvement in this task.

## Figures and Tables

**Figure 1 cancers-14-03648-f001:**
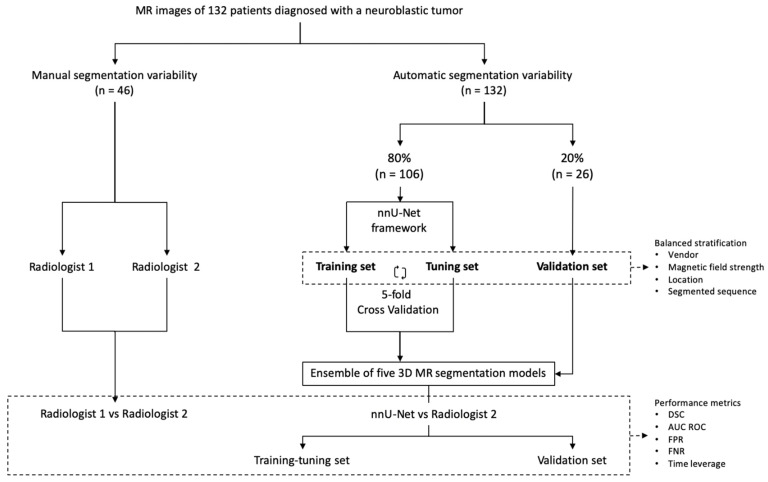
Study design. The first part consisted of manual segmentation variability, comparing the performance of two radiologists (n = 46). The second part included the training and validation of the nnU-Net using 132 cases manually segmented by Radiologist 2. Training-tuning with cross-validation was performed. The 5 resulting segmentation models obtained with the cross-validation method were used as an ensemble solution to test all the cases of the training-tuning (n = 106) and the validation (n = 26) data sets in order to measure training and validation performance independently. The previous split of the cases into balanced groups considering vendor, magnetic field strength, location and segmented sequence was performed.

**Figure 2 cancers-14-03648-f002:**
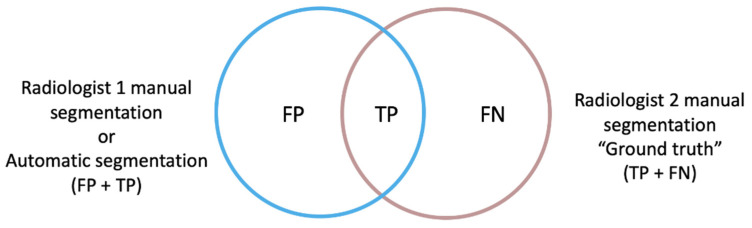
The ground truth (true positive and false negative voxels) corresponds to the manual segmentation performed by Radiologist 2, which was compared firstly to the manual segmentation performed by Radiologist 1 and then to the automatic segmentation obtained by the automatic segmentation model (non-ground truth mask, true positive and false positive voxels). The FPRm considered those voxels that were identified by the model as tumor but corresponded to other structures, divided by the voxels that actually corresponded to the ground truth mask. The FNR measured those voxels belonging to the tumor that the model did not include as such, divided by the ground truth voxels.

**Figure 3 cancers-14-03648-f003:**
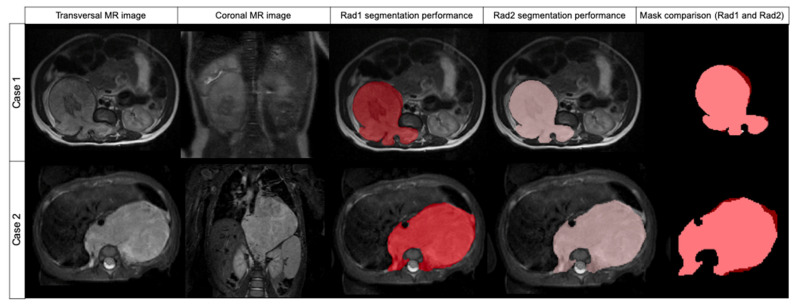
Comparison of two cases segmented by Radiologist 1 (red label) and Radiologist 2 (pink label) and mask superposition and comparison. Case 1 was segmented in T2w while case 2 was segmented in T2w fat-sat. In both cases, DSC was 0.957.

**Figure 4 cancers-14-03648-f004:**
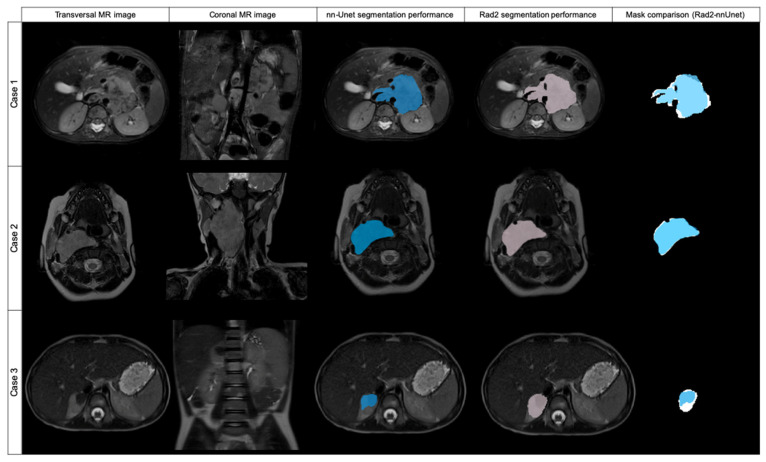
Original transversal and coronal MR images and examples of three cases automatically segmented by nnU-Net (blue labeled) and Radiologist 2 (pink labeled), with mask superposition for comparison. Case 1 was segmented in T2w fat-sat with a DSC of 0.869. Case 2 was segmented on T2w and the DSC obtained was 0.954. Case 3 was segmented with a DSC of 0.617.

**Figure 5 cancers-14-03648-f005:**
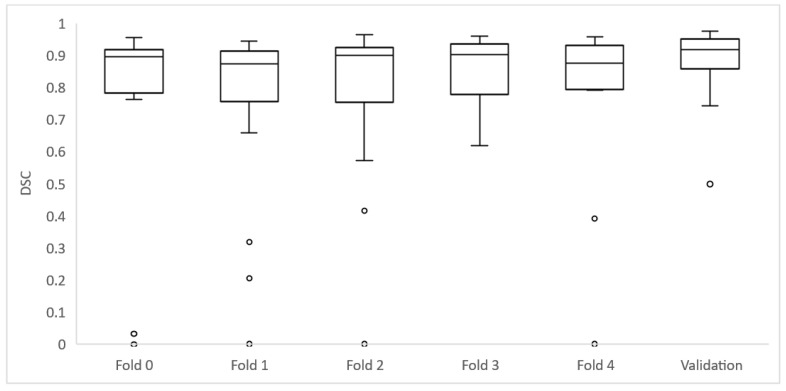
Box plots depicting the whole set of DSC for each fold of the training group and validation set.

**Table 1 cancers-14-03648-t001:** Composition of validation dataset (20% of cases, n = 26), considering four variables for a balanced split: vendor, magnetic field strength, location and segmented sequence.

Validation Set (n = 26)			
Sequence	Equipment	Field Strength	Location
T2	Philips	1.5	Abdominopelvic
T2	Siemens	1.5	Abdominopelvic
T2	Philips	1.5	Abdominopelvic
T2	GE	1.5	Abdominopelvic
T2	GE	1.5	Cervicothoracic
T2	Philips	1.5	Abdominopelvic
T2	Siemens	1.5	Abdominopelvic
T2	Philips	1.5	Cervicothoracic
T2	Siemens	1.5	Abdominopelvic
T2 fat sat	GE	1.5	Abdominopelvic
T2	Siemens	3	Abdominopelvic
T2	GE	1.5	Abdominopelvic
T2	GE	1.5	Cervicothoracic
T2	Philips	1.5	Abdominopelvic
T2 fat sat	Philips	1.5	Abdominopelvic
T2	Siemens	3	Abdominopelvic
T2	Siemens	1.5	Abdominopelvic
T2 fat sat	Siemens	1.5	Abdominopelvic
T2 fat sat	GE	1.5	Cervicothoracic
T2 fat sat	GE	1.5	Abdominopelvic
T2 fat sat	GE	1.5	Abdominopelvic
T2 fat sat	Siemens	1.5	Abdominopelvic
T2	GE	1.5	Cervicothoracic
T2 fat sat	Siemens	1.5	Abdominopelvic
T2 fat sat	Siemens	3	Abdominopelvic
T2 fat sat	GE	1.5	Cervicothoracic

**Table 2 cancers-14-03648-t002:** Inter-observer variability. Performance metrics for inter-observer comparison for manual segmentation, considering DSC, AUC ROC, 1-FPRm and 1-FNR.

	DSC	AUC ROC	1-FPRm	1-FNR
Median	0.969	0.998	0.939	0.998
IQR	0.032	0.004	0.063	0.008
CI	0.042	0.021	0.044	0.042

**Table 3 cancers-14-03648-t003:** Performance metrics for comparison between nnU-Net and Radiologist 2. Cases were divided into 5 folds to perform cross-validation. DSC, AUC ROC, 1-FPRm and 1-FNR for each fold are described.

Fold	Metric	DSC	AUC ROC	1-FPRm	1-FNR
Fold 0	Median	0.895	0.940	0.922	0.882
	IQR	0.121	0.116	0.082	0.233
	CI	0.146	0.117	0.074	0.148
Fold 1	Median	0.873	0.926	0.944	0.856
	IQR	0.110	0.100	0.100	0.100
	CI	0.127	0.066	0.088	0.132
Fold 2	Median	0.899	0.936	0.935	0.875
	IQR	0.131	0.064	0.133	0.133
	CI	0.123	0.062	0.125	0.124
Fold 3	Median	0.901	0.948	0.949	0.897
	IQR	0.122	0.062	0.088	0.124
	CI	0.046	0.030	0.090	0.061
Fold 4	Median	0.874	0.927	0.958	0.856
	IQR	0.134	0.110	0.033	0.221
	CI	0.141	0.071	0.032	0.142

**Table 4 cancers-14-03648-t004:** The 5 resulting segmentation models obtained using the cross-validation method were used as an ensemble solution to test all the cases of the training-tuning (n = 106). Performance metrics for the final results are described. Results are detailed according to location (abdominopelvic or cervicothoracic) and magnetic field strength (1.5 T or 3 T).

	DSC	AUC ROC	1-FPRm	1-FNR
Median	0.965	0.981	0.968	0.963
IQR	0.018	0.010	0.015	0.021
CI	0.031	0.015	0.025	0.031
Cervicothoracic (n = 21)				
Median	0.956	0.975	0.962	0.950
IQR	0.024	0.012	0.015	0.024
CI	0.036	0.018	0.037	0.036
Abdominopelvic (n = 85)				
Median	0.966	0.982	0.969	0.645
IQR	0.015	0.009	0.014	0.019
CI	0.037	0.018	0.030	0.038
1.5 T (n = 93)				
Median	0.965	0.981	0.969	0.963
IQR	0.018	0.011	0.016	0.021
CI	0.029	0.014	0.021	0.029
3 T (n = 13)				
Median	0.964	0.982	0.967	0.964
IQR	0.013	0.005	0.007	0.010
CI	0.145	0.073	0.138	0.145

**Table 5 cancers-14-03648-t005:** Performance metrics for the validation cohort results (n = 26) considering DSC, AUC ROC, 1-FPRm and 1-FNR. Results for Radiologist 2 vs. automatic model are shown. To compare these results to inter-radiologist agreement, Radiologist 1 segmented the 26 cases from the validation dataset and comparisons with Radiologist 2 and to the automatic model were made.

	DSC	AUC ROC	1-FPRm	1-FNR
Radiologist 2 vs. automatic model				
Median	0.918	0.968	0.943	0.938
IQR	0.080	0.051	0.088	0.104
CI	0.059	0.473	0.134	0.063
Radiologist 1 vs. Radiologist 2				
Median	0.920	0.950	0.929	0.930
IQR	0.090	0.192	0.015	0.024
CI	0.038	0.053	0.166	0.058
Radiologist 1 vs. automatic model				
Median	0.915	0.950	0.915	0.912
IQR	0.443	0.122	0.436	0.189
CI	0.114	0.054	0.161	0.104

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
