# Peer review of "Comparative Multicentric Evaluation of Inter-Observer Variability in Manual and Automatic Segmentation of Neuroblastic Tumors in Magnetic Resonance Images"

_cancers, 2022, doi:10.3390/cancers14153648_

Round 1

Reviewer 1 Report

This paper deal with a difficult task in segmentation, the segmentation of neuroblastoma from MRI data. The task that is presented here is challenging for two main reasons. Firstly, neuroblastoma refers to various anatomical locations, which all present some regional specificities due to the proximity of other organs. Secondly, data come from different centers and are acquired with various MR scanners and protocols. This challenge is more conventionally addressed.

To deal with the first challenge, the authors use a stratification of studies distinguishing cervicothoracic studies and abdominopelvic regions both in the training and validation. Despite this point, the diversity of sites of origin (for the tumor) and fields of view should be presented and discussed. In those conditions. I’m not convinced that a stratification according to MR scanner and field strength is the most appropriate (see further comments on tumor volumes and patients’ ages). This point deserves to be discussed.

The main innovation of this paper is the definition of a database of neuroblastoma which is related to the PRIMAGE project. However, cases do come from different trials and local databases, and this database seems to be very heterogeneous, even it should not impact the quality of results. Furthermore, it is necessary to provide the distribution of age and tumor volumes for the different cases of the database. Indeed, the cases with failures should also be discussed according to these variables.

The experimental design has some good points, including:

·      the use of nnU-Net, which seems to be one of the most relevant CNN for this type of data; 

·      the delineation of the tumors by two experienced operators on a significant number of cases (n=46);

·      the metrics that are used to evaluate the performance: besides overlap with “gold standard” metrics, the use of time measurements (for n=20 cases) to obtain an acceptable tumor segmentation is a good point.

However, the experimental design suffers from limitations.

A first point is the definition of the gold standard. The second radiologist is here considered to provide the gold standard segmentation. Of course, it is a practical opportunity, since this second radiologist has performed all the segmentations. But there are no reasons to consider this expert as being superior to the first expert. This point should be discussed. Furthermore, I consider that the so called FPR and FNR metrics should not be used in this paper. Indeed, the discussion between FPR and FNR tends to show that the radiologist 1 has a larger delineation of tumors than radiologist 2. Furthermore, the statement “46 cases were independently segmented by both radiologists after agreement on best tumor definition criteria” needs to be explained. Is the independence condition preserved despite this agreement?

A second point to solve is the absence of link between the 46 cases used for the comparison between the two radiologists and the 20 cases used for the validation database. I would recommend that both radiologists segment the 20 cases of the validation database. Indeed, statistical comparisons would be more powerful if each case of the validation database could be paired with inter radiologist agreement.  

Suggestions for Results

In addition to the previous remark (having inter reader agreement for the 20 cases of the validation database), the AUC of the ROC curve needs to be better defined.

For Figures 3 and 4, a display showing the original T2 image would be useful. In addition, views according to the three planes could also be shown. My suggestion would be to illustrate some cases for which there are some failures.

For tables 2, 3, 4, and 5 choose between (median and IQR) results or (mean and SD) results. Some box plots illustrating the whole set of DSC results would be very helpful.

Cases of failure have to be discussed for the validation database too.

Regarding the stratification of the database (Table 4), some statistical tests should be performed to confirm (infirm) that DSC values are not statistically significant differences between results in the different subgroups.

Similarly, some statistical tests should be performed to study the possible difference in DSC values between the validation and training databases.

Regarding results for time metrics, the comparison between the manual and the fully automatic is not useful. Please remove it. For the time needed for corrections, could you provide some statistical distribution and also add the number of slices which needs some correction.

Finally, the writing of the paper needs to be improved and the paper needs to be condensed, to highlight the most important parts. For instate the subsection analysis and metrics should be reduced, since Dice Score is a very well-known metrics and FPR and FNR are not needed, since they do not appear to have a large impact for this study. Some additional results need to be added. Furthermore, according to the different remarks, the abstract should be reformulated.

Author Response

REVIEWER 1

(The response is uploaded as a word file). 

Q1. This paper deal with a difficult task in segmentation, the segmentation of neuroblastoma from MRI data. The task that is presented here is challenging for two main reasons. Firstly, neuroblastoma refers to various anatomical locations, which all present some regional specificities due to the proximity of other organs. Secondly, data come from different centers and are acquired with various MR scanners and protocols. This challenge is more conventionally addressed. To deal with the first challenge, the authors use a stratification of studies distinguishing cervicothoracic studies and abdominopelvic regions both in the training and validation. Despite this point, the diversity of sites of origin (for the tumor) and fields of view should be presented and discussed.

We appreciate the acknowledgement of the challenging task that is addressed in this manuscript. As it has been pointed out by the reviewer, the dataset accounts for a high heterogeneity in location and data acquisition. As it has been suggested, a more detailed diversity of tumor location has been added to the manuscript, specifying the compartment affected by the tumor (modified on page 3, lines 140-142).

We also agree that the different field of view introduces further heterogeneity and will improve the quality of the manuscript. This has been added (page 4, lines 162-163).

Q2. In those conditions. I’m not convinced that a stratification according to MR scanner and field strength is the most appropriate (see further comments on tumor volumes and patients’ ages). This point deserves to be discussed.

Our multicentric dataset includes patients recruited in 3 different centers and countries (Spain, Italy and Austria), leading neuroblastoma related studies and clinical trials: LINES (European Low and Intermediate Risk Neuroblastoma Protocol clinical trial, leaded by La Fe University and Polytechnic Hospital, Valencia, Spain) and HR-NBL1/SIOPEN (SIOPEN High Risk Neuroblastoma Study, leaded by Children’s Cancer Research Institute, Vienna, Austria, including patients from 12 countries). Therefore, variability was present regarding images acquisition. We aimed to stablish a universal and reproducible model that can be applied on T2 weighted MR images despite the different acquisition scanners and technical parameters. Scanner vendor and field strength have been proved to have significant influence in stripping/segmentation tasks (Souza R, Lucena O, Garrafa J, et al. An open, multi-vendor, multi-field-strength brain MR dataset and analysis of publicly available skull stripping methods agreement. Neuroimage. 2018;170:482-494) and on radiomics features (Ammari S, Pitre-Champagnat S, Dercle L,et al. Influence of magnetic field strength on magnetic resonance imaging radiomics features in brain imaging, an in vitro and in vivo study. Front Oncol. 2021;10:541663). Other authors have pointed out the relevance of the generalization of the model to other manufacturers, stablishing convolutional neural networks for other vendor-neutral segmentation tasks (Jimenez-Pastor A, Alberich-Bayarri A, Lopez-Gonzalez R, et al. Precise whole liver automatic segmentation and quantification of PDFF and R2* on MR images. Eur Radiol. 2021;31(10):7876-7887).

Based on these issues, the vendor and magnetic field stratification approach was designed to provide a reproducible, robust and universal model. We would like to maintain this classification. We have added an explanation to make this point clearer (page 3, line 110; page 4, line 191).

Q3. The main innovation of this paper is the definition of a database of neuroblastoma which is related to the PRIMAGE project. However, cases do come from different trials and local databases, and this database seems to be very heterogeneous, even it should not impact the quality of results. Furthermore, it is necessary to provide the distribution of age and tumor volumes for the different cases of the database. Indeed, the cases with failures should also be discussed according to these variables.

A description of age at first diagnosis (mean, standard deviation and range) is defined in the manuscript. As recommended, information on the age distribution (median and interquartile range) has been added (page 3, line 136). As indicated, a description of tumor volumes has been also added to the manuscript (page 4, line 182-183). The distribution of ages and volumes are also described in the cases of failure, as suggested (page 9, line 337-338).

Q4. The experimental design has some good points, including:

  • The use of nnU-Net, which seems to be one of the most relevant CNN for this type of data; 
  • The delineation of the tumors by two experienced operators on a significant number of cases (n=46);
  • The metrics that are used to evaluate the performance: besides overlap with “gold standard” metrics, the use of time measurements (for n=20 cases) to obtain an acceptable tumor segmentation is a good point.

 However, the experimental design suffers from limitations. A first point is the definition of the gold standard. The second radiologist is here considered to provide the gold standard segmentation. Of course, it is a practical opportunity, since this second radiologist has performed all the segmentations. But there are no reasons to consider this expert as being superior to the first expert. This point should be discussed.

The two observers are experienced pediatric radiologists with previous experience in segmentation tasks, and both could be independently considered as the gold standard. The second radiologist performed all the segmentations and was considered as the ground truth for practical reasons (as net training and validation were based on these segmentations). Manual segmentations were performed by two observers to analyze the variability between them. In the discussion (first paragraph) we approached the issue of interobserver variability in manual segmentation, indicating that the manual ground truth mask may have errors associated to the human-based segmentation methodology. A brief explanation has been added (page 12, line 441-443) to indicate the practical motivations for considering Radiologist 2 as the ground truth.

Q5. Furthermore, I consider that the so called FPR and FNR metrics should not be used in this paper. Indeed, the discussion between FPR and FNR tends to show that the radiologist 1 has a larger delineation of tumors than radiologist 2.

We appreciate the suggestion. One of the strengths of our study was to determinate that the automatic segmentation model achieves a better performance regarding the FPR than the manual variability. This result indicates that the automatic model gets a lower FPR, avoiding the inclusion of non-tumor voxels and reducing this bias in the extraction of quantitative imaging features for the radiomics models. We would like to keep this two spatial overlap-based metrics for the definition of errors in the classification, as we believe that they help to understand the direction of errors when comparing two masks.

Q6. Furthermore, the statement “46 cases were independently segmented by both radiologists after agreement on best tumor definition criteria” needs to be explained. Is the independence condition preserved despite this agreement?

Both radiologists agreed on the segmentation criteria to be used before performing the segmentations. Then, each one performed a blinded segmentation of all the cases independently, without knowing the results of the other radiologist. We have clarified this point in the manuscript to highlight the independence condition (page 4, line 179-180).

Q7. A second point to solve is the absence of link between the 46 cases used for the comparison between the two radiologists and the 20 cases used for the validation database. I would recommend that both radiologists segment the 20 cases of the validation database. Indeed, statistical comparisons would be more powerful if each case of the validation database could be paired with inter radiologist agreement.  

Thanks for this appreciation. As proposed by the reviewer, radiologist 1 performed the manual segmentations of the 26 cases of the validation dataset. We have changed Table 5, adding the results of validation of Radiologist 1 vs Radiologist 2, and Radiologist 1 vs automatic model. (Page 11, line 410-412. Table 5).

Q8. In addition to the previous remark (having inter reader agreement for the 20 cases of the validation database), the AUC of the ROC curve needs to be better defined.

As proposed, an explanation and formula have been added to better define the ROC AUC (page 6, line 240-245).

Q9. For Figures 3 and 4, a display showing the original T2 image would be useful. In addition, views according to the three planes could also be shown.

Figures 3 and 4 have been modified according to this suggestion, adding transversal and coronal images without segmentation masks (pages 8 and 9).

Q10. My suggestion would be to illustrate some cases for which there are some failures.

Figure 4 has been modified, and a third case with a lower DSC (0.617) has been added to illustrate cases with different DSC (0.954; 0.869; 0.617) (page 9).

Q11. For tables 2, 3, 4, and 5 choose between (median and IQR) results or (mean and SD) results.

We have chosen to maintain the median and IQR, as not every variable has a normal distribution. Changes have been made in the manuscript (Tables 2, 3, 4, and 5).

Q12. Some box plots illustrating the whole set of DSC results would be very helpful.

A new figure (Figure 5) is added to depict the whole set of DSC for each fold of the training group and validation sets.

Q13. Cases of failure have to be discussed for the validation database too.

An explanation of failure for the validation cases has been added to the manuscript (pages 10 and 11, lines 380-408).

Q14. Regarding the stratification of the database (Table 4), some statistical tests should be performed to confirm (infirm) that DSC values are not statistically significant differences between results in the different subgroups. Similarly, some statistical tests should be performed to study the possible difference in DSC values between the validation and training databases.

We have performed an ANOVA test for comparing means of DSC attending to the effects of location (abdominopelvic vs. cervicothoracic) and magnetic field strength (1.5 vs. 3T). No differences were obtained, and no interaction between both factors was found. As 7 atypical cases were found, after their removal we applied again the ANOVA test which did not show significative differences, and residues did not present a deviation from normality. There were no differences for DSC mean values for the location and magnetic field factors, and the results were identical after removing atypical values (page 10, lines 361-365).

Q15. Regarding results for time metrics, the comparison between the manual and the fully automatic is not useful. Please remove it.

The main reason for analyzing the time leverage is to demonstrate the huge reduction in time that is achieved with the developed tool, as we believe that this improves the clinician workflow and facilitates the tasks of segmentation. Thus, we would like to highlight the important reduction in the time required for segmentation when using the automatic tool, as it represents an important advantage for radiologist’s clinical practice.

Q16. For the time needed for corrections, could you provide some statistical distribution and also add the number of slices which needs some correction.

As suggested by the reviewer, the number of corrected slices has been added (page 11, line 420 and 427). Also, additional statistical distribution measurements (median, mean and standard deviation) have been added to the manuscript (line 428).

Q17. Finally, the writing of the paper needs to be improved and the paper needs to be condensed, to highlight the most important parts. For instate the subsection analysis and metrics should be reduced, since Dice Score is a very well-known metrics and FPR and FNR are not needed, since they do not appear to have a large impact for this study. Some additional results need to be added. Furthermore, according to the different remarks, the abstract should be reformulated.

We are really grateful for the enriching comments. We have explained why the FPR and FNR should be maintained as metrics in our manuscript, and thus they are still described in the abstract. The writing and grammar were reviewed.  

Reviewer 2 Report

I enjoyed reading this paper which is worthwhile to be published in Cancers after revision. I believe this study provides great insights in the use of deep learning based auto-segmentation model(s) to detect neuroblastic tumors in MR images in clinical settings. I would like to ask authors to provide additional information/data to improve the robustness of this paper.

- I think the case number of validation (test) set is small as compared to the training sets. I believe it would be better to demonstrate the segmentation accuracy of deep learning model using more cases (for example, around the same number as training sets: around 100 cases). Is it possible to add more validation cases for this study?

- I believe it would be better to show the values of confidence interval (CI) in each score (e.g., DSC, AUC ROC....)

- Could you please show the representative false-positive and false-negative prediction for the nnU-Net model?

Author Response

REVIEWER 2

Answers are uploaded as a Word file.

Comments and Suggestions for Authors

Q1. I enjoyed reading this paper which is worthwhile to be published in Cancers after revision. I believe this study provides great insights in the use of deep learning based auto-segmentation model(s) to detect neuroblastic tumors in MR images in clinical settings. I would like to ask authors to provide additional information/data to improve the robustness of this paper. I think the case number of validation (test) set is small as compared to the training sets. I believe it would be better to demonstrate the segmentation accuracy of deep learning model using more cases (for example, around the same number as training sets: around 100 cases). Is it possible to add more validation cases for this study?

We really appreciate the positive feedback received and the suggestions made to improve the manuscript quality. Regarding the number of validation cases, we have divided our dataset according to the rule of 80% for training and 20% for validation, which is commonly used in studies of machine learning and model training. Our scope was to construct by training and tuning the model, and to perform a first internal validation with a 20% of the patients of the dataset. But, as validation is a fundamental step, we are currently recruiting more cases from additional centers. As we do not have these cases now, we would like to maintain the 20% of the cases for validation in this article.

Q2. I believe it would be better to show the values of confidence interval (CI) in each score (e.g., DSC, AUC ROC....)

We have added the CI in each score (tables 2, 3, 4 and 5), as suggested by the reviewer.  In the tables, we have depicted both interquartile range as a measure of dispersion, and CI as a measure of precision (page 6, line 226).

Q3. Could you please show the representative false-positive and false-negative prediction for the nnU-Net model?

The results of FPR and FNR for the final ensemble solution model have been added (page 10, line 360-361) and are shown in Table 4. The explanation of the FPR and FNR metrics are on page 7. The FPR and FNR results are interpreted in the Discussion, to allow a full comprehension of the impact of these metrics in the posterior processes of radiomic extraction. The results of FPR and FNR for model validation are shown in Table 5.

Round 2

Reviewer 1 Report

Authors have implemented most of my comments.

I have two major remarks concerning their revision:

1) If the authors choose to keep what they call ‘FPR' and ‘FNR', they
have to use a different naming at least for FPR (in the abstract and in
the text) because their definition of FPR does not correspond to the
conventional definition (as they note it) and this fact is clearly
misleading for readers

2) In the abstract the time saving is over-optimistic since in the real
life, the reader has to check the quality of the segmentation that is
provided by the automatic approach and to introduce some corrections if
needed. So the time saving should be more realistically evaluated (and
despite this correction, it will remain useful for end-users).

In addition, I have suggested that age and or tumour volume could be
factors explaining possible failures of the method. Could you introduce
a statistical test to check that point ? There is clearly a trend, but
it could be also tested

Author Response

Reviewer 1 - Round 2

Authors have implemented most of my comments. I have two major remarks concerning their revision:

We thank you again for these interesting and enriching corrections that really improve the quality of the manuscript.

Q1) If the authors choose to keep what they call ‘FPR' and ‘FNR', they have to use a different naming at least for FPR (in the abstract and in the text) because their definition of FPR does not correspond to the conventional definition (as they note it) and this fact is clearly misleading for readers

We appreciate this suggestion, as we consider this a very important point. As indicated by the reviewer, the FPR used in our manuscript differs from the FPR conventionally used in literature. To indicate this difference, we have changed the name of the FPR to “modified version of FPR” (FPRm) (page 1, line 33; page 7, line 274), and the abbreviation FPR has been modified to FPRm to indicate this difference.

Q2) In the abstract the time saving is over-optimistic since in the real life, the reader has to check the quality of the segmentation that is provided by the automatic approach and to introduce some corrections if needed. So the time saving should be more realistically evaluated (and despite this correction, it will remain useful for end-users).

Thank you for this appreciation. We have modified the methods (page 7, 305) and the results to better explain the process of visual validation and/or manual edition (page 12, 439-442).

We have also added a brief explanation of these point in the discussion (page 13, line 506-511), explaining, as suggested by the reviewer, that the result of the automatic tool regarding time leverage is over-optimistic, as in practice the reader has to visually validate the quality of all the segmentations provided by the automatic tool before introducing some corrections, if needed. Besides, we have modified the abstract (page 1, line 42-43) indicating the time reduction after manual adjustment.

Q3) In addition, I have suggested that age and or tumour volume could be factors explaining possible failures of the method. Could you introduce a statistical test to check that point ? There is clearly a trend, but it could be also tested

As suggested by the reviewer, we introduced age and volume as corrective factors in the evaluation of the effects of location and magnetic field in the DICE. A generalized linear model was performed on the DICE response variable, considering the location and magnetic field variables as categorical factors and age and tumor volume as covariates. The results show that no significant effect is observed in the DICE of any of the variables (p-value= 0.052 for age and 0.169 for volume). Therefore, the effects of location and magnetic field, as well as their interaction, continue to be insignificant when the correction for age and volume is introduced (page 10, 374-379).

Reviewer 2 Report

Thank you.

Author Response

We thank you again for the interesting corrections that really improve the quality of the manuscript.